# Hepatocyte Nuclear Factor 4α (HNF4α) Plays a Controlling Role in Expression of the Retinoic Acid Receptor β (*RARβ*) Gene in Hepatocytes

**DOI:** 10.3390/ijms24108608

**Published:** 2023-05-11

**Authors:** Reza Zolfaghari, Jessica A. Bonzo, Frank J. Gonzalez, A. Catharine Ross

**Affiliations:** 1Department of Nutritional Sciences, Pennsylvania State University, University Park, PA 16802, USA; rxz7@psu.edu; 2Laboratory of Metabolism, Center for Cancer Research, National Cancer Institute, National Institutes of Health, Bethesda, MD 20814, USA

**Keywords:** HNF4α, *RARβ* gene promoter, retinoic acid, retinoic acid receptors, HNF4α ligand binding domain, retinoic acid response element, DNA binding site

## Abstract

HNF4α, a member of the nuclear receptor superfamily, regulates the genes involved in lipid and glucose metabolism. The expression of the *RARβ* gene in the liver of HNF4α knock-out mice was higher versus wildtype controls, whereas oppositely, *RARβ* promoter activity was 50% reduced by the overexpression of HNF4α in HepG2 cells, and treatment with retinoic acid (RA), a major metabolite of vitamin A, increased *RARβ* promoter activity 15-fold. The human *RARβ2* promoter contains two DR5 and one DR8 binding motifs, as RA response elements (RARE) proximal to the transcription start site. While DR5 RARE1 was previously reported to be responsive to RARs but not to other nuclear receptors, we show here that mutation in DR5 RARE2 suppresses the promoter response to HNF4α and RARα/RXRα. Mutational analysis of ligand-binding pocket amino acids shown to be critical for fatty acid (FA) binding indicated that RA may interfere with interactions of FA carboxylic acid headgroups with side chains of S190 and R235, and the aliphatic group with I355. These results could explain the partial suppression of HNF4α transcriptional activation toward gene promoters that lack RARE, including *APOC3* and *CYP2C9,* while conversely, HNF4α may bind to RARE sequences in the promoter of the genes such as *CYP26A1* and *RARβ*, activating these genes in the presence of RA. Thus, RA could act as either an antagonist towards HNF4α in genes lacking RAREs, or as an agonist for RARE-containing genes. Overall, RA may interfere with the function of HNF4α and deregulate HNF4α targets genes, including the genes important for lipid and glucose metabolism.

## 1. Introduction

There is increasing interest in how micronutrients, such as vitamin A, play roles in the regulation of macronutrient metabolism. The major metabolite of vitamin A, all-trans-retinoic acid (RA), is known to regulate many physiological processes through its function as an activating ligand, or sometimes a repressive ligand, for nuclear retinoic acid receptors (RAR) including RARα, RARβ, and RARγ, each of which partner with retinoid X receptors (RXRα, RXRβ, and RXRγ), forming dimeric complexes that bind specifically to retinoic acid response elements (RARE) present in the promoter of target genes [1,2,3,4]. The canonical RARE consists of a core of two hexameric motifs of RGKTCA (where R and K represent any purine or pyrimidine, respectively), which are most often oriented as a direct repeat (DR) spaced by two and five nucleotides, although spacings of zero, one, and eight are also known, as well as inverted repeats (IR) separated by zero, three, and nine nucleotides [4]. The RAR-RXR heterodimer is capable of binding in vivo to a wider variety of half-site spacings compared with other nuclear receptors, including the thyroid hormone receptor, vitamin D receptor, and peroxisome proliferator-activated receptors, which each heterodimerize with RXR and then bind to DR4, DR3, and DR1 response elements, respectively [4]. In most RA-responsive genes, RARE are located upstream of the transcription start site (TSS) [5,6,7]. Despite over three decades of research, only a small number of the many genes reported to be physiologically responsive to retinoid treatment in vivo have been definitively shown to be controlled in a direct manner by the transcriptional mechanism involving the direct binding of ligand-activated RAR-RXR to cognate RARE. In fact, it appears that the majority of RA-responsive genes may be regulated indirectly [6]. Therefore, elucidating alternative mechanisms is important for understanding the full potential of RA-regulated gene expression in vivo.

Among the directly controlled genes targeted by RA for transcriptional activation is the *RARβ* gene itself [7,8]. In earlier reports, a DR5 RARE was identified in the proximate region upstream of the TSS and was shown to be responsive to both RARα and RARβ, but was not significantly regulated by other nuclear receptors [7]. In earlier studies in vivo, we and others demonstrated that genes encoding RARα, RARβ, and RXRα are significantly expressed in the liver [9,10,11], the major organ for vitamin A storage and the distribution of retinol to the plasma for delivery to the entire body. *RARβ* mRNA levels are significantly regulated by dietary vitamin A and by RA itself in the liver of adult rats, as the expression of *RARβ* mRNA was about 50 to 70% lower in vitamin-A-deficient rats compared with the vitamin-A-sufficient controls [9,11]. Previous research showed that the administration of RA [11] or retinol [10] was effective at up-regulating the expression of *RARβ* mRNA in rats with low vitamin A status. This positive autoregulation of RARβ by retinoid compounds could be a means of elevating the level of RARβ receptor protein to, in turn, amplify the effect of RA on its target genes. Among the probable target genes for RARβ is *CYP26A1*, a member of the CYP26 subfamily of cytochromes P450 (CYP) enzymes, which specifically encodes a CYP that catalyzes the oxidative hydroxylation of RA, resulting in its inactivation [12]. The *CYP26A1* gene is highly responsive to the presence of RA, as shown by the elevated expression of *CYP26A1* mRNA during embryonic development, where CYP26A1 controls RA concentration in specific regions, as well as in the adult liver and cultured liver cells [13,14,15,16]. Several functional RARE elements exist in the *CYP26A1* gene promoter, including one RARE located proximal to the TSS through which RARs bind and regulate the RA-mediated induction of *CYP26A1* [17,18]. 

Another important area of research concerns whether and how nuclear receptors interact to facilitate or attenuate gene expression. For CYP26A1, although RARs play significant roles in its induction, other nuclear transcription factor(s) may also play significant roles. One of these may be hepatocyte nuclear factor 4α (HNF4α), a highly conserved member of the nuclear receptor superfamily of ligand-dependent transcriptional factors [19]. Previously, we showed that HNF4α binds to the RARE site present in the region proximal to the transcription start site in the *CYP26A1* promoter and increases C*YP26A1* mRNA expression in HepG2 cells following treatment with RA [20]. HNF4α is expressed at high levels in the liver, small intestine, and kidney and at lower levels in the pancreas and colon [21]. HNF4α is a key regulator of numerous hepatocyte genes that play important roles in lipid and glucose metabolism and in the catabolism of xenobiotics and drugs, particularly by members of the CYP family [22,23,24]. Similar to other members of the nuclear receptor superfamily, HFN4α contains distinct DNA-binding and ligand-binding domains. Originally, HNF4α was considered to be an orphan member of the nuclear receptor superfamily because, unlike most other members, HNF4α activates transcription in the absence of exogenously added ligands. Later, a crystallography study reported that fatty acids are embedded in the protein’s structure and may serve as endogenous HNF4α ligands [25]. Specifically, linoleic acid (LA), an essential long-chain fatty acid, was later identified as a reversible physiological ligand for HNF4α; however, ligand occupancy was not shown to have a significant effect on the HNF4α transcriptional activity [19]. HNF4α is associated with several human diseases including diabetes, hemophilia, hepatitis, atherosclerosis, and inflammatory bowel diseases [26]. 

As complete elimination of the *Hnf4a* gene has been shown to result in embryonic lethality, we initiated this study to investigate the impact of acute loss of HNF4α in mouse livers on the expression of the *Cyp26a1* gene by employing a temporally inducible hepatocyte-specific *Hnf4a*-null mouse line [27,28]. While we initially hypothesized that *Cyp26a1* gene expression would be reduced in the livers of *Hnf4a*-null mice, contrary to our expectation, we found that the expression of the *Cyp26a1* gene is elevated higher in the absence of hepatic HNF4α in mice treated with RA compared with that in RA-treated *Hnf4a^F/F^* littermate controls. This suggests that there may be complex interactions between HNF4α and retinoid signaling. Therefore, in the present study, we explored the role of HNF4α further, finding evidence that RA may act both as an agonist and as an antagonist of the HNF4α receptor in a manner dependent on the presence or absence of RARE sites in the promoter of target genes.

## 2. Results

### 2.1. Loss of HNF4α Induces Expression of the RARβ Gene in Mouse Liver

Previously, we showed that HNF4α increased the promoter activity of the human *Cyp26a1* gene in HepG2 cells upon treatment with RA [20]. Based on those results, we investigated whether HNF4α has any effect on expression of the *Cyp26a1* gene in the liver of intact mice. *Hnf4a^F/F^* (control) and *Hnf4a^F/F;AlbERT2cre^* mice, with an inducible deletion of HNF4α in the liver, were treated orally with either oil as a vehicle or with RA for 6 h, after which the animals were euthanized for the collection of liver tissue. The total RNA was extracted from individual liver samples and analyzed for individual genes by qRT-PCR using 18S ribosomal RNA as the internal control. No *Hnf4a* mRNA was detected in the liver samples from *Hnf4a^F/F;AlbERT2cre^* mice treated with either the vehicle or RA (Figure 1A). In *Hnf4a^F/F^* mice (controls), *Hnf4a* mRNA increased by 50% after RA treatment compared with the vehicle-treated *Hnf4a*^fl/fl^ mice (Figure 1A). The expression of HNF4α was previously shown to be increased in hepatocytes treated with RA [20,29,30]. 

To check the integrity of the *Hnf4a^F/F;AlbERT2cre^* mouse line, we measured the mRNA level of several genes as positive controls, including *ApoC3* (Figure 1B) and several others (Appendix A and Appendix A, genes for Perp-pending, Rarres2, Rdh5, Rdhe2, Saa4, and RBP4). *ApoC3* mRNA expression has been shown to be increased by HNF4α in hepatocytes [31]. We found that the lack of HNF4α in the liver resulted in the suppression of the *ApoC3* gene expression by about 80% (Figure 1B). The treatment of mice with RA reduced, although not significantly, the mRNA levels in the liver compared with that in the mice treated with oil as the vehicle control. 

*Cyp26a1* gene expression is normally very low or undetectable in the livers of mice fed vitamin-A-adequate diets [15,16]. In this study, we confirmed that the *Cyp26a1* gene expression was very low in the livers of the vehicle-treated control mice (Figure 1C). The expression levels were slightly higher in the vehicle-treated *Hnf4a^F/F;AlbERT2cre^* mice compared with the *Hnf4a^F/F^* mice (Figure 1C). The treatment of mice with RA increased *Cyp26a1* mRNA several fold in both *Hnf4a^F/F^* and *Hnf4a^F/F;AlbERT2cre^* mice. Contrary to our initial expectation, we observed higher levels of *Cyp26a1* mRNA in the livers of *Hnf4a^F/F;AlbERT2cre^* mice than in the *Hnf4a^F/F^* mice (Figure 1C). 

Holloway et al. (2008) published a microarray study that showed that the expression level of *RARβ,* but not other *RAR* mRNAs, was about 2.5-fold higher in the liver of *Alb-Hnf4a*^-/-^ than those in the *Hnf4a^F/F^* group (in both male and female mice). We found that the average mRNA level of *RARβ* was about five-fold higher in the liver of *Hnf4a^F/F;AlbERT2cre^* mice compared with *Hnf4a^F/F^* mice (Figure 1D). Retinoic acid treatment increased *RARβ* mRNA levels in the livers of *Hnf4a^F/F;AlbERT2cre^* mice as well as in the *Hnf4a*^fl/fl^ mice (Figure 1D). The knockdown of HNF4α in the liver caused an increase in the expression of RARβ, which resulted in a higher expression of the CYP26A1 gene when the animals were treated with RA. 

### 2.2. Overexpression of HNF4α Up-Regulates Promoter Activity of the Human RARβ Gene in HepG2 Cells Treated with RA

HepG2 cells, a well-differentiated cell model for human hepatocytes [32], have been used to determine the expression and promoter analysis of a number of genes including RARs and RA-induced genes. Therefore, we first treated HepG2 cells with 1 µM RA for 4 h and 24 h and then collected the cells for RNA extraction and RNA seq analysis. While the *RARβ* gene expression was induced 19-fold after 4 h and more than 50-fold after 24 h of treatment with RA, *RARα* and *RARγ* mRNA increased less than 50% at those two time points. As anticipated, HepG2 cells did not express endogenous *CYP26A1* mRNA prior to retinoid treatment; however, upon treatment of the cells with RA, the *CYP26A1* mRNA levels were increased by more than 500-fold after 4 h and by more than 5000-fold after 24 h. These results demonstrate both that the beta isoform of RAR is most sensitive to RA, and that the CYP26A1 gene is highly inducible by RA, providing a good model for further studies of the impact of HNF4a on the expression of these genes. 

Having shown that HepG2 cells could be an appropriate model to analyze the function of hepatic *RAR* genes in response to RA, we tested whether HNF4α could act on the promoter of the *RARB* gene. For this, we used a cloned fragment of the human *RARβ* gene spanning from −1.7 kbp from TSS to +0.217 kpb of the 5’UTR region [33,34] as the driving promoter on a *p*GL3-Basic-luc vector used as the reporter gene. HepG2 cells were co-transfected with the *RARβ* promoter construct and the *p*RLTK plasmid together with either human HNF4α, the combination of RARα and RXRα (referred to below as RARα/RXRα), or all three transcription factors, for 24 h, after which the cells were treated further with either the vehicle or RA for 1 to 24 h prior to measurement of the luciferase activity. After 24 h of transfection, the cells overexpressing RARα/RXRα exhibited a 50% increase in *RARβ* promoter activity (Figure 2A). In contrast, transfection with HNF4α either in the presence or absence of RARα/RXRα reduced the promoter activity of *RARβ* by more than 50%. Upon further incubation of the cells without RA for 1 to 24 h following the transfection period, HNF4α, either alone or with RARα/RXRα, still suppressed the *RAR*β promoter activity (Figure 2A). However, upon incubation of the cells with RA during this post-transfection period, HNF4α either alone or with RARα/RXRα increased the promoter activity, with maximum induction after 8 h (Figure 2B). We also found a similar interaction between human HNF4α and human RARα/RXRα when testing the activity of the promoter of the mouse *RARβ* gene in the presence or absence of RA (Figure 2C). 

Whether acting directly or indirectly, HNF4α appears to control the expression of the *RARβ* gene. While the lack of HNF4α in the liver resulted in an increased expression of the *RARβ* gene (Figure 1D), the overexpression of HNF4α suppressed the promoter activity of the *RARβ* gene in HepG2 cells (Figure 2A). However, upon treatment of the cells with RA, HNF4α induced the promoter of the *RARβ* gene significantly (Figure 2B,C). 

### 2.3. HNF4α Response Element(s) May Reside within the RARE Sites in the Promoter of the RARβ Gene

In order to learn where HNF4α binds in the promoter region of the human *RARβ* gene, we first analyzed the sequence of the full-length (FL) promoter using the MatInspector computer program (www.genomatix.de accessed on 4 March 2017). In addition to the two DR5 RARE that are present proximal to TSS, we identified three putative HNF4α binding sites within this 1.7 kbp promoter region, as illustrated in Figure 3A, for the FL promoter. One site lies close to the proximal region (from −255 to −267 bp) and the other two sites lie in the middle region of the promoter (−633 to −645 and −1083 to −1107 bp). To identify regions of the promoter that are responsive to HNF4α, we conducted sequential deletions from the 5’ end of the FL construct through the use of either restriction enzymes or through PCR amplification, as shown in Figure 3A. The individual clones with these deletions as well as the WT FL clone, each along with *p*RLTK as an internal control, were co-transfected with either HNF4α, RARα/RXRα, or all three transcription factors, and then following transfection, the cells were treated with either vehicle or RA for 24 h. Compared with the FL construct (construct 1, as the control), elimination of the 5’-end of the promoter region containing the putative HNF4α binding sites did not significantly impact the ability of HNF4α or RARα/RXRα to activate the *RARβ* promoter (Figure 3B). However, elimination of the 5’-end of the FL promoter by SpeI restriction enzyme (construct 4, −89 bp) resulted in a significant reduction in promoter activation through not only HNF4α, but also through RARα/RXRα, even though the RARE sites were intact. In fact, the basal activity of the promoter was almost nil in the cells containing construct 4, either without or with RA (Figure 3B). Further elimination of the 5’-end by SmaI restriction enzyme (−59 bp) resulted in the complete inability of HNF4α to activate the *RARβ* promoter, although residual activity of RARα/RXRα was observed in the same cells treated with RA (Figure 3B). This proximal region contained the previously described RARE1 [7,8], which may account for the minimal activity remaining. These results showed that the upstream region of the promoter extending from −59 bp to −267 bp is not only essential for basal promoter activity, but is also essential for the receptor actions of HNF4α and RARα/RXRα on the *RARβ* promoter in cells treated with RA (Figure 3B).

### 2.4. At Least Two RARE Sites Are Present in the Proximate Region of the RARβ2 Promoter 

Based on these results, we aligned the sequences of the *RAR*β promoter extending from 5’-UTR to about −1700 bp upstream of TSS from the human, mouse, and rat genes using the ClustalW2 program. The sequences for all 3 species had high homology from the 5’-UTR to −167 bp from TSS (Figure 4A). We identified that there are two DR5 RARE sites present in this region (RARE1 and RARE2), and in addition, there is one DR8 site. Interestingly, the downstream half DR of the DR8 site is the upstream half DR of the RARE2 site (Figure 4A). In an earlier report, the RARE1 site had been tested [7,8] and was reported to be responsive to both RARα and RARβ, but not significantly responsive to other nuclear receptors known at the time [7]. To examine the RARE2 and DR8 sites, we disrupted this region with two different clusters of mutations: one mutation was introduced to alter the nucleotides comprising the 5-nucleotide spacer in RARE2 and the other mutation to alter both the DR5 sequences within RARE2 (see Figure 4B for mutated sequences). Then, the FL mutated constructs were co-transfected with either HNF4α, RARα/RXRα, or all three transcription factors into HepG2 cells, which were treated with either the vehicle or RA for 24 h. Compared with the WT FL promoter, mutation of the spacer in RARE2 did not have any significant effect on the action of the receptors; however, the basal activity was more than doubled in the cells treated with RA (Figure 4C). These results were observed consistently in repeated experiments. Mutation of the RARE2 DR sites did not have any significant effect on basal promoter activity in the presence or absence of RA, but it reduced the effect of HNFα on the promoter by more than 95% in the cells treated with RA (Figure 4C). Minimal promoter activity was observed in the cells co-transfected with RARα/RXRα and treated with RA. Thus, the RARE2 site together with the intact DR8 site (Figure 4D) are not only essential for the actions of HNF4α, but also of RARα/RXRα on the *RARβ* promoter in HepG2 cells treated with RA (Figure 4C).

### 2.5. Retinoic Acid but Not Linoleic Acid Induces the Transcriptional Activation of HNF4α toward the RARβ Promoter

As noted earlier, the essential fatty acid linoleic acid is considered as an embedded physiological ligand of HNF4α [19,25]. To test whether LA activates the *RARβ* promoter through HNF4α, we treated HepG2 cells, following transfection with either WT human FL *RARβ* promoter or mutant promoters, both with or without the HNF4α expression vector, with either vehicle, 100 µM LA, or 1 µM RA for 24 h, after which the cells were assayed for *RARβ* promoter activity (Figure 4D). We observed that LA, with or without FBS, had no effect on the activity of the *RARβ* promoter through HNF4α action. In comparison, however, RA activated the promoter through HNF4α by about 15-fold (Figure 4D).

### 2.6. Mutation of the Amino Acid Residues Essential for Fatty Acid Binding in the Ligand Binding Domain of HNF4α Results in Suppression of Its Transcriptional Activation

To test whether RA may have any effect on the transcriptional activation of HNF4α, we used a previously described mutational model [35] for defining the critical residues in the ligand binding pocket of rat HNF4α. Aggelidou et al. showed that point mutations of the residues that come into contact with the fatty acid ligand resulted in a dramatic decrease in transcriptional activity toward the *APOC3* promoter, without affecting the protein expression, binding of HNF4α to DNA, or dimerization of HNF4α. Their results defined the importance of residues S181, M182, L219, L220, R226, and I-346, which line the ligand-binding pocket of rat HNF4α, and of residue I-338, in maintaining the structural integrity in the region, which significantly impairs the transcriptional activity of HNF4α [35]. Residue R212 of rat HNF4α was also mutated on the notion that this residue may be involved in stabilizing the ligand-binding pocket. However, the R212G mutation was found to maintain transcriptional activation potential in their study [35].

As human HNF4α is conserved and highly homologous to rat HNF4α (Figure 5) and both proteins have been shown to associate spontaneously with endogenous fatty acids [25], we used human HNF4α for the mutation analysis. We mutated all corresponding amino acid residues in the ligand binding domain of human HNF4α that correspond to those in the rat protein (Figure 5). These include S190, M191, R221, L228, L229, R235, I-347, and I-355 (Figure 5). Based on crystallographic data for human HNF4α [25], residues S190 and R235 both have direct contact with the carboxyl group of fatty acid, whereas M191, L228, L229, and I-355 interact with the fatty acid’s aliphatic chain. Moreover, R221 may be involved in stabilizing the ligand binding pocket and I-347 has been predicted to maintain the structural integrity of the ligand-binding pocket of human HNF4α.

We then tested the transcriptional activity of the individual mutated HNF4α clones compared with that of the WT protein on the promoters of four different human genes in HepG2 cells. These promoters include the *APOC*3 and *CYP2C9* genes, both of which lack RARE but nevertheless respond to RA, and the *RARβ* and *CYP26A1* genes, which, as noted, contain multiple RARE sites [17,18] (Figure 4). Before evaluating the transcriptional activity of the mutated human HNF4α clones, we examined how the human *APOC3* and *CYP2C9* promoters respond to RA, even though these promoters lack any RARE. For this experiment, HepG2 cells were co-transfected with the individual promoter constructs in *p*GL3-Basic-luc plasmid vectors together with *p*RLTK plasmid containing Renilla-luc as the control reporter, and with either WT human HNF4α, RARα/RXRα, or all three transcription factors. Following transfection, the cells were treated with either the vehicle or 1 µM RA for 24 h, after which the cells were assayed for luciferase activity. HNF4α activated the *APOC3* promoter by about four-fold and the *CYP2C9* promoter by more than 100-fold in HepG2 cells treated with the vehicle (Figure 6A,D). However, RA treatment of the cells reduced the HNF4α-mediated upregulation of the *APOC3* promoter by about 38% and that of *CYP2C9* by more than 90% (Figure 6A,D). The addition of RA and/or RARα/RXRα in the cells not transfected with exogenous HNF4α also reduced the activity of the promoters of both genes in these cells due to the endogenous expression of HNF4α in HepG2 cells [20].

In order to test the transcriptional activity of mutated human HNF4α, HepG2 cells were co-transfected with individual promoter constructs in *p*GL3-Basic-luc plasmid vector together with *p*RLTK plasmid containing Renilla-luc and with individual mutated HNF4α clones. Following transfection, the cells were treated with either vehicle or 1 µM RA for 24 h, then assayed for luciferase activity. Mutation of R221 (Mutant #3), which was predicted to be involved in the stability of the ligand binding pocket, had no effect on the activity of HNF4α (Mutant #3, R221 in Figure 6B,E). However, while partial activity was retained with the mutation of either S190 (Mutant #1) or R235 (Mutant #6), there was loss of transcriptional activity with mutations of the individual residues M191 (Mutant #2), L228 (Mutant # 4), L229 (Mutant #5), I-347 (Mutant #7), and I-355 (Mutant # 8). Transfection with these mutants each reduced the basal activity of the promoters, suggesting that they suppressed endogenous HNF4α activity and acted as dominant negative mutants, possibly through heterodimerization with endogenous HNF4α molecules (Figure 6C,F). This suppression of endogenous HNF4α may suggest that the mutants affect the activation function of the receptor, without interfering with its ability to bind the target sequence in the promoters [35]. Similar results have been reported for transcriptional activation of the rat HNF4α mutants on the *APOC3* promoter [35].

As the promoters of the *RARβ* and *CYP26A1* genes both contain RAREs, they act differently from the promoters of the *APOC3* and *CYP2C9* genes upon co-transfection with HNF4α and treatment of the cells with RA following transfection. While HNF4α activates the promoters of *APOC3* and *CYP2C9* genes, their activation by HNF4α is suppressed in cells treated with RA. In contrast, HNF4α activates the promoters of *RARβ* and *CYP26A1* genes in HepG2 cells treated with RA. Mutation of the residues S190 (Mutant #1) and R235 (Mutant #6), each of which has a direct interaction with the carboxyl group of the endogenous fatty acid, resulted in a loss of more than 80% of the transcriptional activity of HNF4α toward the *APOC3* and *CYP2C9* gene promoters, while most of its transcriptional activity was retained on the promoters of *RARβ* and *CYP26A1*, compared with the activity of WT HNF4α (Figure 7A,B). Mutation #3 of the R221 residue, involved in stabilizing the ligand-binding pocket, had little to no effect on transcriptional activity toward the *RARβ* and *CYP26A1* promoters (Figure 7A,B), similar to *APOC3* and *CYP2C9* genes (Figure 6B,E). Mutation of I-347 (Mutant #7), a residue predicted to maintain the structural integrity of the ligand-binding pocket of HNF4α, resulted in complete loss of its activity toward not only the *APOC*3 and *CYP2C9* genes (Figure 6B,E), but also the *CYP26A1* and *RARβ* genes (Figure 7A,B, respectively). The mutation of four residues, including M191 (Mutant #2), L228 (Mutant #4), L229 (Mutant # 5), and I-355 (Mutant #8), each of which has a direct interaction with the aliphatic chain of fatty acids, resulted in complete loss of HNF4α transcriptional activity toward the *APOC3* and *CYP2C9* gene promoters (Figure 6B,E). Except for residue I-355 (Mutant #8), mutation of the other three amino acid residues resulted in the loss of HNF4α activity toward the *CYP26A1* and *RARβ* gene promoters (Figure 7A,B, respectively).

## 3. Discussion

In this study, we first observed that the acute loss of HNF4α resulted in an increase in *RARβ* gene expression in the livers of adult mice. As anticipated, RARβ was induced by RA in the livers of the control mice, and we also observed induction by RA in the livers of mice lacking HNF4α. Thus, HNF4α is not required for the response to RA. We next examined whether *Cyp26a1,* as a probable *RARβ* target gene, induced by RA, was greater in the livers of mice lacking HNF4α. To better understand whether and how HNF4α regulates the expression of the *RARβ* gene, we used the *RARβ* promoter extending from +217 bp to −1700 bp from TSS in reporter gene constructs expressed in HepG2 cells, a cell line in which RARβ is naturally expressed and highly regulated by RA. We found that the overexpression of HNF4α in the absence of exogenous RA resulted in the suppression of the *RARβ* promoter in HepG2 cells; however, treatment of the cells with RA following transfection with HNF4α resulted in activation of the promoter of *RARβ*. Apparently, HNF4α has dual controlling effects over the expression of the *RARβ* gene, either directly or indirectly. On the one hand, HNF4α exerts suppressive action over the expression of the *RARβ* gene, which may be through its promoter, and, on the other hand, HNF4α may activate the *RARβ* promoter upon treatment of the cells with RA. 

The results from the DNA sequence analysis of the human *RARβ* gene indicated that there are at least three putative HNF4α response elements present throughout the promoter. We showed that elimination of these elements did not have any significant effect on the transactivation activity of HNF4α toward the *RARβ* promoter. In fact, the minimum sequence requirement for HNF4α activation of the *RARβ* promoter extended to about −267 upstream of TSS. This region (a) is downstream of the putative but non-functional HNF4α response elements; (b) is highly conserved between human, mouse, and rat species; and (c) contains at least two DR5 RAREs, namely RARE1 and RARE2 together with a DR8, which is also believed to be responsive to retinoic acid receptors. RARE1 has been reported to be responsive to RAREs, but not to other hormone nuclear receptors [7,8]. We found that RARE2, a DR5, is functionally active in response to not only RARα/RXRα, but also to HNF4α. Mutation of the DR5 nucleotides resulted in complete loss of the promoter activity in response to HNF4α and RARα/RXRα. Mutation of the spacer nucleotide sequence did not have any significant effect on the promoter in response to the receptors, but did increase the basal activity of the promoter in cells treated with RA. Thus, the DR5 element RARE2 and possibly with DR8 are the sites through which HNF4α activates the promoter of *RARβ* in hepatocytes following treatment with RA. 

HNF4α is considered one of the most promiscuous DNA binders among the nuclear receptors and among transcription factors in general [4]. Although it has been found to occupy aGt CAAAGt Ca sites as a general consensus element [36], HNF4α has been reported to regulate genes in the liver through RARE sites, namely DR2 [37,38] and DR5 [20], present in the promoters. For example, erythropoietin, a cytokine promoting progenitor cell proliferation, is required for erythropoietic differentiation. The *EPO* gene has been shown to be a direct transcriptional target gene of RA signaling during early erythropoiesis (prior to embryonic day E12.5) in the fetal mouse liver [38]. The promoter of the erythropoietin gene contains a functional RARE that is occupied by RXRα/RARα and activated by RA during E9.5–E11.5 [38]. After E11.5, *EPO* expression is dominated by HNF4α through the same RARE. As another example, hepatic glucokinase catalyzes the phosphorylation of glucose to glucose 6-phosphate, a step that is essential for glucose metabolism in the liver, as well as for the induction of glycolytic and lipogenic genes. The promoter of the glucokinase gene (*Gck*) has been shown to contain a functional RARE site that interacts with not only retinoic acid receptors but also with HNF4α and chicken ovalbumin upstream promoter transcription factor II (COUP-TFII) in rat primary hepatocytes, to integrate vitamin A and insulin signaling [37]. Previously, we also showed that HNF4α binds to the RARE site present in the proximal region of the promoter of *CYP26A1* and increases the expression of the gene in HepG2 cells following treatment with RA [20]. 

In addition to possessing a DNA binding domain, HNF4α, similar to the other members of the hormone nuclear receptor superfamily, contains a distinct ligand binding domain. However, unlike the hormone nuclear receptors, HNF4α apparently does not need any added exogenous ligand for its function as a transcriptional activator. The crystal structure of the HNF4α ligand binding domain revealed that the protein, which adopts a canonical fold, is present in two conformational states, closed and open forms, within each homodimer [25]. Although the protein was reported to be crystallized without an added ligand, the ligand binding pockets of both the closed and open conformations were found to contain fatty acids [25]. In fact, a fatty acid was reported to be an ideally suited ligand for HNF4α [25]. The carboxylic acid headgroup of the fatty acid ion pairs with the guanidinium side-chain group of Arg226 of the rat protein to fix the orientation, while the aliphatic portion of the fatty acid occupies a long narrow pocket that is lined with hydrophobic side chains of the amino acid residues [25]. Homologous to critical residues present in the rat HNF4α ligand binding pocket [35], we found that R235, S190, M191, L228, L229, and I355, participating in the formation of the ligand binding pocket of human HNF4α, play an important role in the transcriptional activation of HNF4α toward the promoters of human *APOC3* and *CYP2C9* genes. Whereas the carboxylic acid headgroup of the fatty acid interacts with side-chain groups of Arg235 and of S190, the aliphatic portion of the fatty acid occupies a long narrow pocket that is lined with hydrophobic side chains of the amino acid residues, including M235, L228, L229, and I355. Moreover, similar to the R212 residue in rats, HNF4α mutation of R221, which has been speculated to be involved in stabilizing the ligand binding pocket [35], had no effect on the transcriptional activation of human HNF4α. On the other hand, I347, similar to the I338 residue in rat protein, which was reported to maintain the structural integrity of the ligand-binding pocket of HNF4α [35], was shown to be important in human HNF4α transcriptional activity toward the promoters tested. 

HNF4α has been shown to activate both *APOC3* [31] and *CYP2C9* [39] genes in hepatocytes. Here, we used the promoters of these genes as controls and found that HNF4α activates the promoter of both genes in HepG2 cells. Although the promoters of both genes lack RAREs, HNF4α transcriptional activation was suppressed toward those genes when the cells were treated with RA. In contrast, the promoters of the *RARβ* and *CYP26A1* genes, each of which contain multiple RAREs, could be activated by HNF4α only when the cells were treated with RA following transfection. We found mutation of the critical residues in the ligand binding pocket of HNF4α, namely S190 and Arg235, which are involved in the interaction with the carboxyl group of fatty acids, did not have any significant effect on the HNF4α transcriptional activation toward the promoters of *RARβ* and *CYP26A1* genes in HepG2 cells treated with RA. However, mutation of M235, L228, and L229, which coordinate with the aliphatic portion of the fatty acid, were shown to be important in human HNF4α transcriptional activity toward the promoters of both *CYP26A1* and *RARβ* genes. In addition, I347, which is involved in maintaining the structural integrity of the ligand-binding pocket of HNF4α [35], was shown to play an important role in human HNF4α transcriptional activity toward the promoters of those genes. Based on these results, RA may interfere with the interaction of residues, namely R235 and I355, both in closed conformation only, with endogenous fatty acid ligands. Therefore, RA may act as an antagonist toward HNF4α transcriptional activity on the promoters of genes lacking RAREs but activated by HNF4α. On the other hand, RA may act as an agonist for HNF4α transactivation of gene promoters possessing RARE sites. 

In summary, we showed that (a) loss of HNF4α resulted in the induction of the *RAR*β gene expression in the livers of adult mice; (b) overexpression of HNF4α in HepG2 cells suppressed the *RARβ* gene promoter, but upregulated the promoter when the cells were treated with RA; and (c) HNF4α may act through at least one of the two functional RAREs present in the promoter of the *RARβ* gene in the region proximal to its transcription start site. As fatty acids have been reported as endogenous ligands for HNF4α, our results imply that RA may interfere either directly or indirectly with the interaction of these fatty acids with the amino acid side chains of critical residues in the HNF4α ligand binding domain and, as a result, RA may suppress the genes activated by HNF4α that contain no RAREs, while being able to activate those genes possessing RAREs through HNF4α. In either case, excess vitamin A, including RA itself, may interfere in the transactivation function of HNF4α and thereby deregulate the genes involved in lipid and glucose metabolism, as well as xenobiotic metabolism. As a result, excess vitamin A may contribute to dyslipidemia, diabetes, and metabolic syndrome, abnormalities for which HNF4α has been shown by others to be a significant factor. Because HNF4α is involved in several critical macronutrient pathways, and RA is the major functional metabolite of vitamin A as well as being a clinically important drug, recognizing the intersection of these factors helps to deepen our understanding of the complex ways in which hepatic lipid and carbohydrate metabolism are regulated.

## 4. Materials and Methods

### 4.1. Materials

Human hepatoma HepG2 cells were obtained from the American Type Culture Collection (ATCC, Manassas, VA, USA) and maintained in Eagle’s Minimum Essential Medium (EMEM) supplemented with 10% fetal bovine serum (FBS) and 0.5% penicillin−streptomycin (1x)at 37 °C in a 5% CO_2_−air incubator. The cells were plated and used at 60 to 80% confluency. RA was purchased from Sigma-Aldrich Chemical Company (St. Louis, MO, USA) and prepared as stocks at 1 and 10 mM in ethanol. *p*GL3-Basic-luc, *p*RLTK, and *p*GEMT-Easy plasmid vectors were purchased from Promega (Madison, WI, USA) and *p*cDNA3.1 plasmid vector used for expression was from Invitrogen (Carlsbad, CA, USA). MMLV Reverse transcriptase, RNase inhibitor, oligo dT nucleotide, dNTP, and T4 polynucleotide kinase were from Promega (Madison, WI, USA), while High Fidelity Tag DNA polymerase, Lipofectamine 2000 transfection reagent, and Trizol reagent were all from Invitrogen. Plasmid DNA Purification Kits and the DNA gel extraction kit were from Omega Bio-Tek (Thermo Scientific, Waltham, MA, USA). All of the restriction enzymes were obtained from New England Biolab (Ipswich, MA, USA). For the preparation of all other reagents in our laboratory, we followed the protocols reported in [40].

### 4.2. Animal Experiment and RNA Analysis 

The hepatocyte-specific *Hnf4a*-null mice (*Hnf4a^F/F;AlbERT2cre^*) were described previously [27]. Animal experiments were performed in accordance with Institute of Laboratory Animal Resources guidelines and protocols approved by the Animal Care and Use Committee of the National Cancer Institute. In brief, *Hnf4a^F/F^* and *Hnf4a^F/F;AlbERT2cre^* male mice were fed a diet containing tamoxifen (1g/kg) for three days and then returned to regular rodent chow for 3 days prior to treatment with either vehicle corn oil (0.1 mL/mouse) or RA in oil (1 mg/kg, 0.25 mg/mL) administered by oral gavage. Six hours after dose administration, the animals were euthanized and tissues were harvested and snap-frozen in liquid nitrogen. 

For our studies, the total RNA was extracted from individual liver samples using Trizol reagent, dissolved in DEPC-treated deionized water, and then analyzed both quantitatively and qualitatively by Nanodrop spectrophotometry (Thermo Scientific, Waltham, MA, USA). The individual RNA samples were run in denatured agarose gel by electrophoresis in order to check the quality of the RNA. For mRNA expression analysis, 2 µg of total RNA sample was first reverse transcribed in a 25 µL reaction and then diluted with autoclaved deionized water to 150 µL. The, 5 µL of the diluted sample was used to quantify the gene expression level by real time qPCR with SYBR Green containing reaction mix (BioRad, Hercules, CA, USA) using gene specific primer pairs, as shown in Table 1 and Appendix A. The PCR program was set to run first at 95 °C for 3 min for activation of the polymerase and then for 40 cycles each of 20 s at 95 °C for melting, and 1 min at 60 °C for annealing and extension. Specificity of the primer pairs was confirmed by performing the melting curves and also by running the PCR products in agarose gel by electrophoresis after the PCR.

### 4.3. Analysis of RAR Gene Expression in Cultured Cells

HepG2 cells were grown in EMEM with 10% fetal bovine serum and 1x penicillin−streptomycin in six-well plates until they reached to about 70 to 80% full confluency. The cells were then treated with RA at a final concentration of 1 µM for 0, 4, and 24 h, each in four repeats, after which the cells were washed with 1 × PBS and then subjected to RNA extraction with the Trizol reagent. The total RNA samples were cleaned further using the method reported previously [41]. The RNA in the samples was quantified, tested for quality, and then submitted to the Nucleic Acid Facility for RNA Seq analysis, as previously described [42].

### 4.4. Cloning and Preparation of Vector Constructs

Construction of the plasmid vectors including *p*GL3-Basic-hCYP26A1-E4-luc (submitted to addgene.org), *p*GL3-Basic-hRARβp-luc (human RARβ2 promoter), *p*GL3-Basic-hCYP2C9p-luc, *p*cDNNA3.1-hRARα.hRXRα (submitted to addgene.org), and *p*cDNA3.1-hHNF4α were reported previously [18,20,33,34,43]. The full-length HNF4α mutants were constructed by PCR-mediated mutagenesis, using appropriate primers (Table 2) and the WT human HNF4α cDNA as the template. To construct the desirable mutants, the internal primers presented in Table 2 were used along with the WT amino-terminal primer, 5′ TTGGATCCGCCACCATGCGACTCTCCAAAACCC-3′, and WT carboxyl-terminal primer, 5′-TTTCTAGACTAGATAACTTCCTGCTTGG-3′. The PCR-amplified fragments were cloned into the vector *p*cDNA 3.1(+) at the BamHI and XbaI sites. All of the mutants were submitted for DNA sequencing analysis for verification at the Nucleic Acid Sequencing Facility of the Pennsylvania State University. For cloning the promoter of the *RARβ* mouse gene, a 1.9 kbp fragment spanning from −1918 bp to −17 bp upstream of the ATG codon was amplified by PCR from mouse genomic DNA using the primer pair shown in Table 1, and high fidelity *Taq* DNA polymerase following the protocol recommended by the manufacturer. Similarly, a fragment of the promoter of human *ApoC3* gene spanning from −1420 bp to +36 bp from the TSS was constructed by PCR from human genomic DNA using the primer pair shown in Table 1. The cycling program was 94 °C, 2 min for initial denaturation, and 40 cycles of 94 °C, 15 s; 55 °C, 30 s; and 68 °C, 2 min. The amplified products were run on agarose gel and the DNA band was cut from the gel and extracted using a DNA extraction kit. The DNA was first cloned into *p*GEMT-Easy plasmid vector by TA cloning and sub cloned into *p*GL3-Basic luciferase as previously described for other gene promoters [44,45]. The DNA expression plasmid was subjected to sequencing for confirmation, as above.

### 4.5. Transfection and Luciferase Assay Analysis

HepG2 cells grown in EMEM with 10% fetal bovine serum and 1 x penicillin/streptomycin in T-75 flask were trypsinized and transferred into 24-well plates 1–2 days before transfection and grown to about 60 to 70% confluency. The cells were then transfected for 24 h with a total of 0.8 µg plasmid DNA and 2 µL of Lipofectamine 2000 per well in an EMEM medium containing 3% fetal bovine serum with no antibody following the protocol recommended for Lipofectamine usage. In a typical transfection experiment using two expression vectors, the plasmid DNA included 0.6 µg *p*GL3-Basic-luc plasmid vector containing the promoter construct and *p*RLTK plasmid containing the Renilla-luc gene for transfection efficiency (7:1 w:w for DNA) and two expression vectors of 0.1 µg for each plasmid DNA. An equivalent amount of empty vector DNA (*p*cDNA3.1^+^), as the control, was used to fill the DNA for expression vectors. Following transfection, the cells were incubated with full growth medium containing 10% fetal bovine serum and 1 µM RA in ethanol at a final concentration of 0.001% ethanol. After incubation at 37 °C for up to 24 h, the cells were washed with PBS and then lysed to assay for firefly-luc and Renilla-luc activities using the DRL luciferase assay system from Promega in Luminator 20. Promoter activity was defined as the ratio of firefly-luc to Renilla-luc activity.

### 4.6. Statistical Analysis

Each reported activity is the average of at least three wells with standard deviation of the mean. Data from animal experiments are reported as standard error of the mean. Where indicated, the data were analyzed by one-way ANOVA followed by Fisher’s least significant difference test using Prism 9 Statistical Software (GraphPad, San Diego, CA, USA). A value of *p* < 0.05 was considered statistically significant.

## Figures and Tables

**Figure 1 ijms-24-08608-f001:**
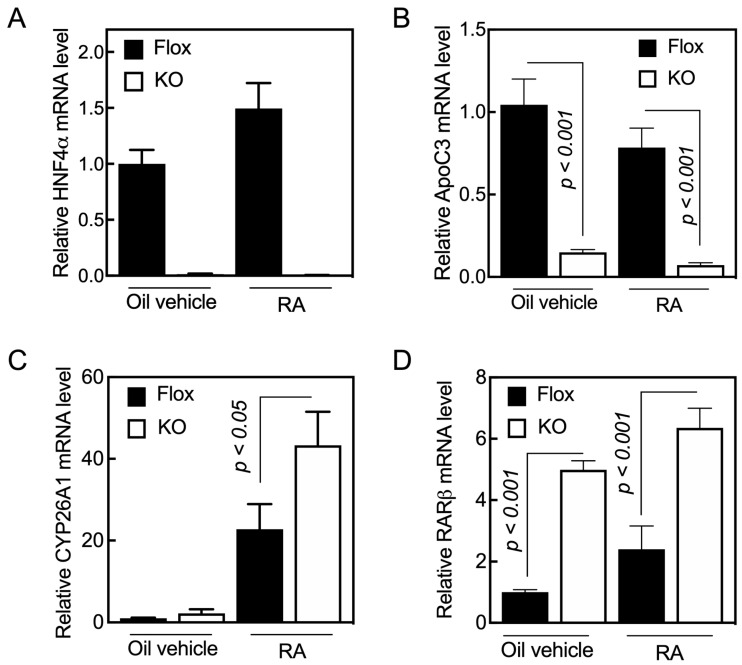
Acute loss of HNF4α results in the induction of *RARβ* gene expression in the livers of adult mice. *Hnf4a^F/F^* (Flox) and *Hnf4a^F/F;AlbERT2cre^* (KO) mice were administered either oil as the vehicle control (0.1 mL/mouse) or RA in oil (0.25 mg/mL, 0.1 mL/mouse) for 6 h, and then euthanized for collection of the liver tissue samples. The total RNA was extracted from tissue samples for qPCR analysis of gene expression using gene specific primers (Table 1) for (**A**) *HNF4**α***, (**B**) *ApoC3*, (**C**) *Cyp26A1*, and (**D**) *RARβ*. The relative mRNA expression of the specific gene over 18S rRNA level, as the internal control, was set as 1 in the *Hnf4a^F/F^* group (Flox) treated with oil as the vehicle. Data represent the mean ± SEM of *n* = 3 to 5 mice per group.

**Figure 2 ijms-24-08608-f002:**
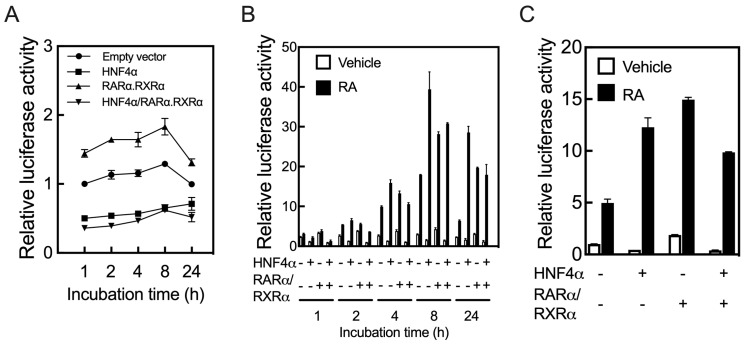
HNF4α regulates the promoter of the *RARβ* gene in HepG2 cells. HepG2 cells in 24-well plates were co-transfected for 24 h with a fragment of the human *RARβ* promoter expanding from −1.7 kbp from transcription start site to +0.217 kpb, as the full length promoter construct, in *p*GL3-basic-luc vector together with *p*RLTK plasmid containing Renilla-luc, as the control, and with either human HNF4α, RARα/RXRα, or all three transcription factors, and then treated further either without (**A**) or with RA for 1 to 24 h, after which the cells were collected to assay for luciferase activity. HNF4α alone or with RARα/RXRα suppressed the promoter activity of the human *RARβ* gene in the cells treated without RA (**A**), but upregulated the promoter when the cells were treated with RA following transfection (**B**). HNF4α regulates the promoter of the mouse *RARβ* gene in HepG2 cells (**C**). It suppresses the promoter activity of the mouse gene (empty bars in C) in HepG2 cells treated with the vehicle, but it increases the promoter activity in the cells treated with RA (black bar in C).

**Figure 3 ijms-24-08608-f003:**
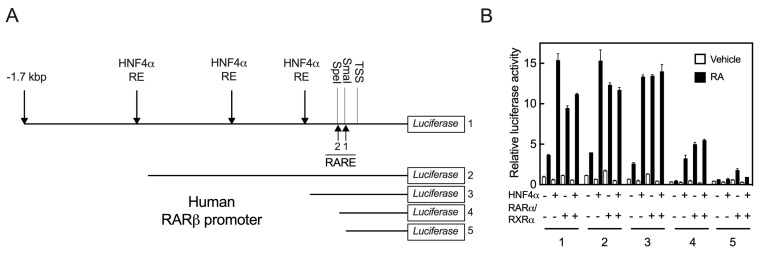
HNF4α response may reside within the RARE sites in the promoter of *RARβ* gene. (**A**) Sequence of the full length (FL) promoter of the human *RARβ* gene (construct # 1) was analyzed by using the MatInspector computer program (www.genomatix.de accessed on 4 March 2017) for the presence of HNF4α response elements and retinoic acid response elements (RARE). To identify regions of the promoter responsive to HNF4α, sequential deletions from the 5’ end of the FL construct were made by either restriction enzymes or by PCR (Construct #’s 2 to 5). (**B**) The individual deleted clones as well as the WT FL clone, each with *p*RLTK as the internal control, were co-transfected with either HNF4α, RARα/RXRα, or all three transcription factors. Following transfection, the cells were treated with either vehicle or RA for 24 h, after which the cells were washed and lysed for measurement of the luciferase activity, as described in the Section 4.

**Figure 4 ijms-24-08608-f004:**
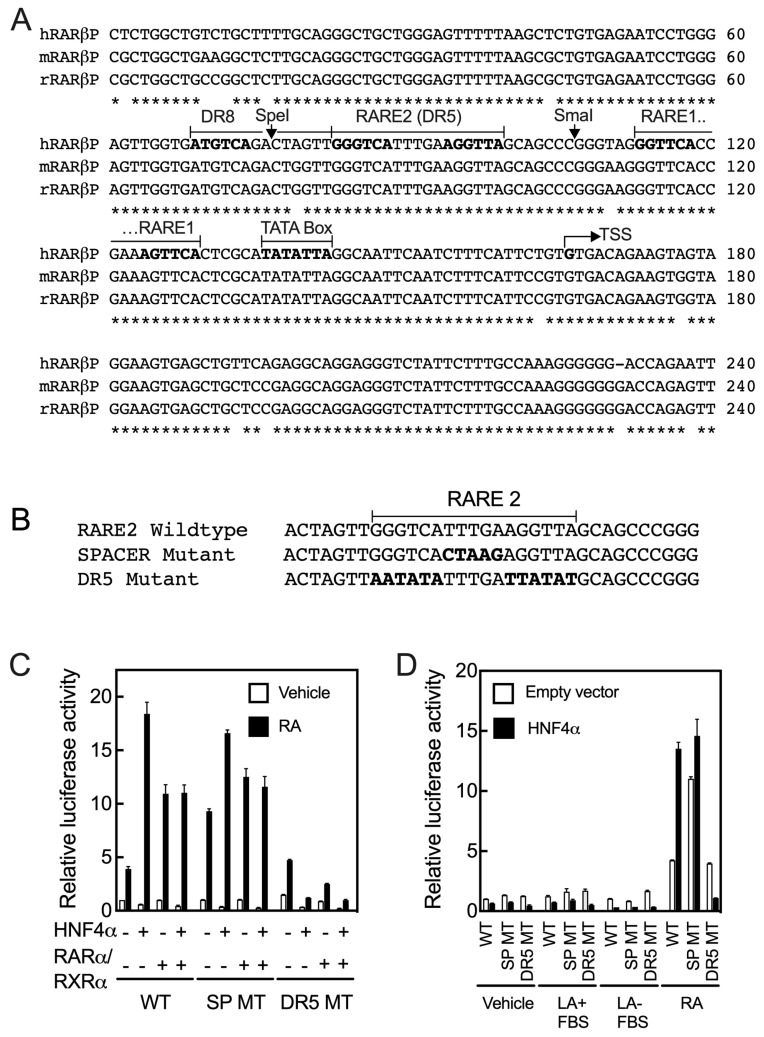
Two putative DR5 RARE sites with an DR8 are present in the proximal region to the transcription start in the promoter of the *RARβ* gene. (**A**) Sequences of the proximal region in *RARβ* promoters are highly conserved in humans, mice, and rats. There are two direct repeat 5 (DR5) and one DR8 as potential RARE sites present in this region. Asterisks (*) indicate identity and bold type shows the hexameric sequences. (**B**) Sequences of WT and mutant RARE2 constructs. (**C**) Mutation of the RARE2 DR5 elements but not that of the spacer in the full-length promoter clone reduced the activity of HNF4α on the *RARβ* promoter by more than 95% in the cells treated with RA. (**D**) Exogenous added RA (1 µM) but not linoleic acid (100 µM) in either the absence or presence of 10% fetal bovine serum activates the WT promoter of the *RARβ* gene in HepG2 cells. Data from each bar represent the mean of *n* = 3 wells ± SD. RARE, retinoic acid response element; DR, direct repeat; WT, wildtype; SP, spacer; MT, mutant; LA, linoleic acid; FBS, fetal bovine serum.

**Figure 5 ijms-24-08608-f005:**
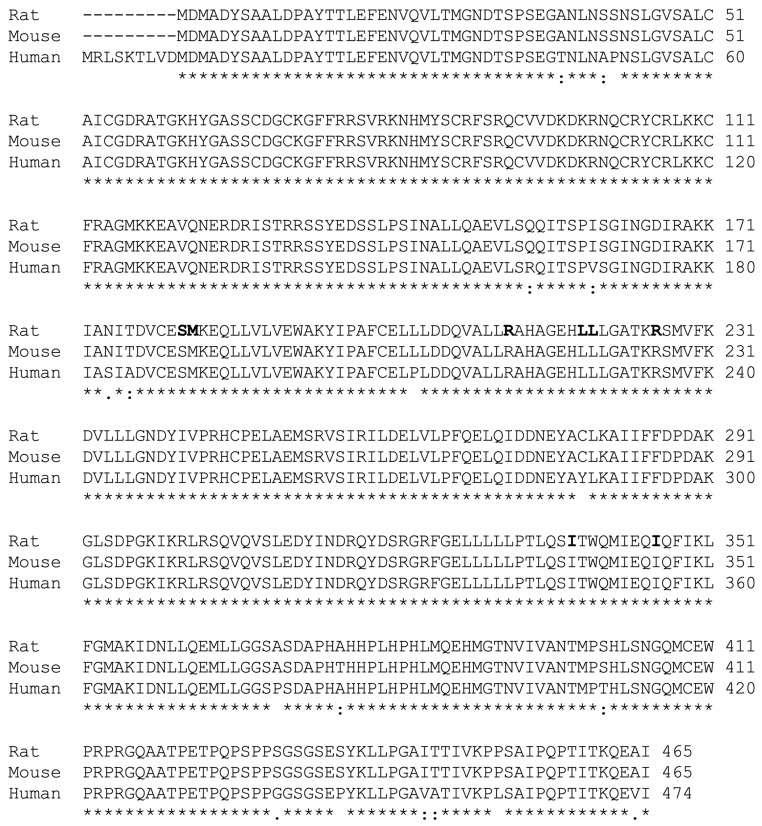
HNF4α is highly conserved among rats, mice, and humans. Alignment of the amino acid sequence of the rat (accession # BAA01411) and mouse (accession # EDL06337) HNF4α protein with that of the cloned human homologue (Zolfaghari and Ross, 2014). The mutated residues based on the model reported for the ligand binding domain of rat HNF4α [35] are highlighted. Asterisks (*) indicate identify and bold type shows the hexameric sequences.

**Figure 6 ijms-24-08608-f006:**
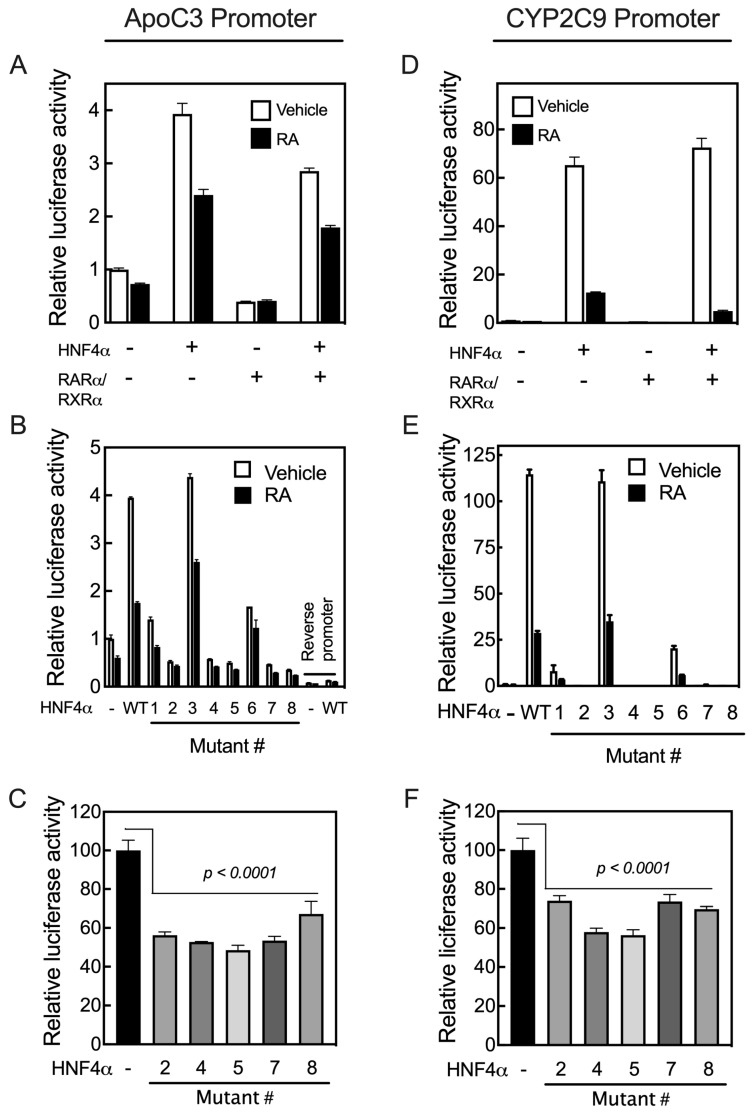
Mutation of the critical residues present in the ligand binding domain of human HNF4α suppresses transcription activation of the promoters of human *APOC3* and *CYP2C9* genes. HepG2 cells were co-transfected with either the *p*GL3-b-hApoC3 (**A**–**C**) or *p*GL3-b-hCYP2C9 (**D**–**F**) promoter construct together with either hHNF4α, RARα/RXRα, or all three transcription factors, and then treated with either vehicle or 1 µM RA for 24 h, after which the cells were assayed for their luciferase activity. The effects of the individual mutant residues in the ligand binding domain of the human HNF4α compared with WT HNF4α on the promoter activity of *APOC3* (**B**) or *CYP2C9* (**E**) in HepG2 cells treated with either vehicle or 1 μM RA for 24 h. The endogenous HNF4α transcriptional activity toward *APOC3* (**C**) and *CYP2C9* (**F**) promoters was assessed in HepG2 cells with or without the addition of human HNF4α mutants. Data from each bar represent the mean of *n* = 3 wells ± SD. HNF4α mutant #’s are as follows: (1) S190K, (2) 191MK, (3) R221G, (4) L228K, (5) L229K, (6) R235G, (7) I347K, and (8) I355K.

**Figure 7 ijms-24-08608-f007:**
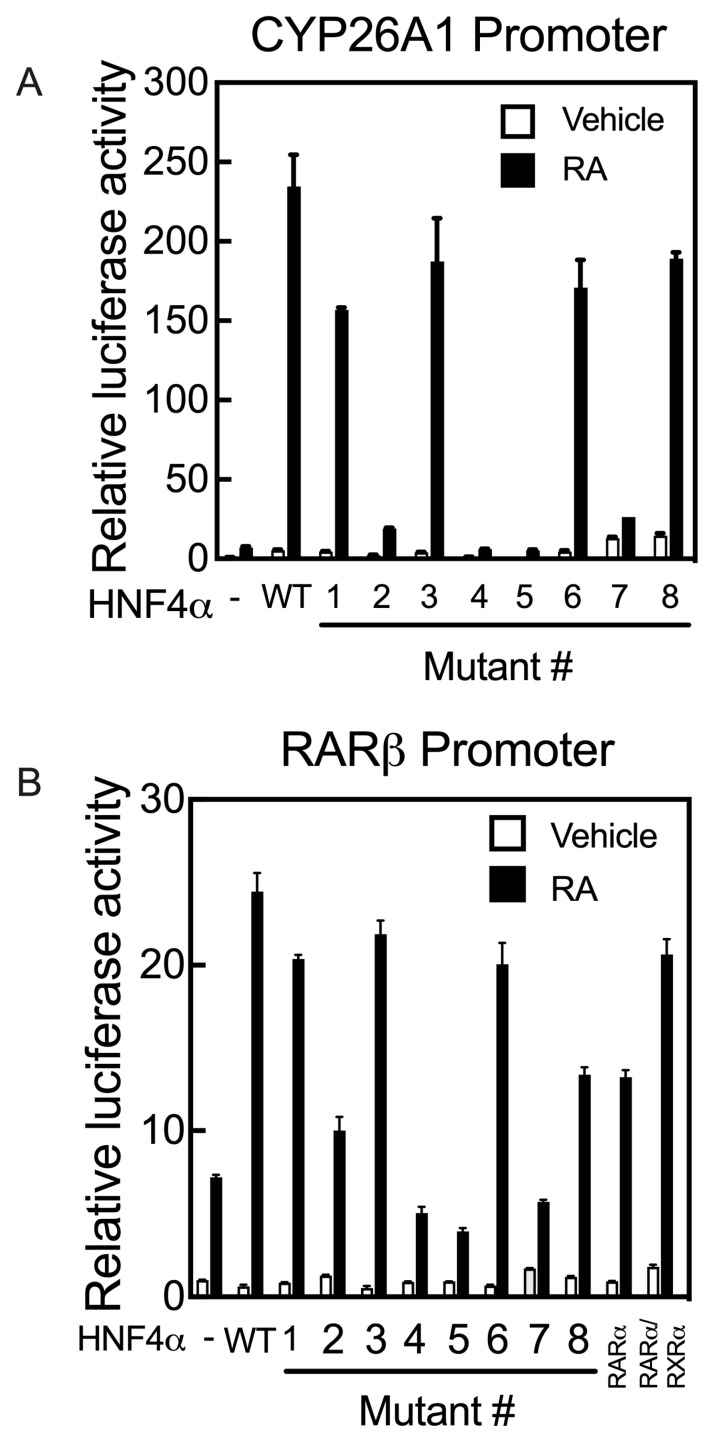
Mutation of the critical amino acid residues present in the ligand binding domain of human HNF4α suppresses transcription activation of the promoters of human *CYP26A1* and *RARβ* genes in HepG2 cells treated with RA. HepG2 cells grown in 24-well plates were co-transfected with either *p*GL3-b-hCYP26A1 (**A**) or *p*GL3-b-hRARβ (**B**) promoters, each with *p*RLTK as the control, together with either wildtype (WT) HNF4α or its individual mutants (Mutant # 1 to 8), and then treated with either vehicle or 1 µM RA for 24 h, after which the cells were assayed for luciferase activity. Data from each bar represent the mean of *n* = 3 wells ± SD. HNF4α mutant #’s are as follows: (1) S190K, (2) 191MK, (3) R221G, (4) L228K, (5) L229K, (6) R235G, (7) I347K, and (8) I355K.

**Table 1 ijms-24-08608-t001:** List of primers for RT-PCR analysis and cloning.

Gene Name	Analysis	Primer Pairs
Mouse HNF4α	RT-PCR	S: ACATCCCGGCCTTCTTCTGCGAAC A: CATTGCCTAGGAGCAGCACGTCCT
Mouse ApoC3	RT-PCR	S: ACATGGAACAAGCCTCCAAG A: GGAGGGGTGAAGACATGAGA
Mouse CYP26A1	RT-PCR	S: TGCAGGCACTAAAACAATCG A: TCAATCGCAGGGTCTCCTTA
Mouse RARβ	RT-PCR	S: GCCTTCTCAGTGCCATCTGT A: CTGTTCTCCACTGAGCTGGG
Mouse RARβ	RT-PCR	S: CTTCCTCCTCCTCGGGTGTA A: GGCTTTGAGCAGGGTGATCT
Mouse RARβ	RT-PCR	S: TTCATGTTCGGGGCTGGG A: GGTAGCCCGATGACTTGTCC
18S Ribosomal RNA	RT-PCR	S: CGCGGTTCTATTTTGTTGGT A: AGTCGGCATCGTTTATGGTC
Mouse RARβ promoter	Cloning	F: TAATTGGACAGGGGTGGTCT R: CGCTCTGCAAAAGTGCTTATC
Human ApoC3 promoter	Cloning	F: tgctagcCTTGGAGCCCTTAGAGCCTT R: tctcgagTACCTGGAGCAGCTGCCTC
Human RARβ mutant promoter	Cloning	F: ACTAGTTAATATATTTGATTATATGCAGCCCGGGTAGGGTTC R: GATCCCAAGTTCTCCTTCCA
Human RARβ mutant promoter	Cloning	F: ACTAGTTGGGTCACTAAGAGGTTAGCAGCCCGGGTAG R: ACTAGTGATTGATCCCAAG
Human RARβ deleted promoter	Cloning	F: TTCTAGTCACAGCTCTGAGC R: GATCCCAAGTTCTCCTTCCA
Human RARβ deleted promoter	Cloning	F: AAGCACTTCTTGTATTGTTT R: GATCCCAAGTTCTCCTTCCA

Lower case letters indicate the restriction sites for cloning. S: sense; A: antisense; F: forward; R: reverse.

**Table 2 ijms-24-08608-t002:** List of primers used for the construction of mutant HNF4a clones.

Mutant #	Mutant Residues	Primer Sequence
1	S190K	S: TTAAGCTTAAGGAGCAGCTGCTGGTTC A: TTAAGCTTCTCACACACATCTGCGATGC
2	M191K	S: TTAAGCTTGAGCAGCTGCTGGTTCTC A: TTAAGCTTGGACTCACACACATCTGCG
3	R221G	S: TTGGATCCCATGCTGGCGAGCACCTGC A: TTGGATCCGAGCAGGGCCACCTGGTCG
4	L228K	S: TTAAGCTTCTCGGAGCCACCAAGAGATC A: TTAAGCTTGTGCTCGCCAGCATGGGCTC
5	L229K	S: TTAAGCTTGGAGCCACCAAGAGATCC A: TTAAGCTTCAGGTGCTCGCCAGCATGG
6	R235G	S: TTGGATCCATGGTGTTCAAGGACG A: TTGGATCCCTTGGTGGCTCCGAGC
7	I347K	S: TTAAGCTTTGGCAGATGATCGAGCAG A: TTAAGCTTGCTCTGCAAGGTGGGCAGC
8	I355K	S: TTAAGCTTTTCATCAAGCTCTTCGGC A: TTAAGCTTCTGCTCGATCATCTGCCAG

S: sense; A: antisense.

## Data Availability

The data that support the findings of this study are available from the corresponding author upon reasonable request. Several of the constructs used in this study have been deposited with and are available from AddGene (addgene.org).

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
