# Peer review of "Hepatocyte Nuclear Factor 4α (HNF4α) Plays a Controlling Role in Expression of the Retinoic Acid Receptor β (RARβ) Gene in Hepatocytes"

_ijms, 2023, doi:10.3390/ijms24108608_

Round 1

Reviewer 1 Report

Zolfaghari et al. present a well thought out and described manuscript where the relationship between RXR beta and HNF alpha is highlighted. The use of the mutants was very helpful in illustrating the relationship between all the components. 

Introduction - describes the relationship between the nuclear factors very well and their relationship with each other.

Results

Please be cautious in the description of figure one describing non-significant results.

Last sentence of section 2.2 is very long and has a lot of information included, please consider breaking this sentence up.

Section 2.6 seems to no longer be presented as a header.

Discussion

Line 433 - something is grammatically causing this sentence to read strangely. 

The conclusion(s) appear appropriate based on the results.

Methods

Line 587 - needs a space before "1"

Language didn't appear to be an issue here. Noticed a few inconsistent use of commas and italics (i.e., in vivo) but nothing major.

Author Response

Comments and Suggestions for Authors

Zolfaghari et al. present a well thought out and described manuscript where the relationship between RXR beta and HNF alpha is highlighted. The use of the mutants was very helpful in illustrating the relationship between all the components. 

Response: Thank you for this positive and helpful review. We have addressed the specific points below.

Introduction - describes the relationship between the nuclear factors very well and their relationship with each other.

Results

Please be cautious in the description of figure one describing non-significant results.

Response: We have rewritten this sentence (line159).

Last sentence of section 2.2 is very long and has a lot of information included, please consider breaking this sentence up.

Response: We have broken that sentence into two—thank you for this suggestion.

Section 2.6 seems to no longer be presented as a header.

Response: We have changed it to a header. Thank you!

Discussion

Line 433 - something is grammatically causing this sentence to read strangely. 

Response: We have edited this sentence, now in line # 441.

The conclusion(s) appear appropriate based on the results.

Methods

Line 587 - needs a space before "1"

Response: We have corrected that. Now it is line # 596. Thank you!

Comments on the Quality of English Language

Language didn't appear to be an issue here. Noticed a few inconsistent use of commas and italics (i.e., in vivo) but nothing major.

Response: We have corrected these inconsistencies.

Reviewer 2 Report

The authors tested the expression of Cyp26a1 gene in the liver specific Hnf4a KO mice. Contrary to their earlier in vitro observation, Cyp26a1 mRNA level was increased in the HNF4a deleted liver especially after RA treatment. Upregulation of a nuclear hormone receptor Rarb by HNF4a deletion or RA treatment was attributed to the controversial Cyp26a1 gene expression by HNF4a. Therefore, the authors launched various in vitro analyses including reporter gene assays along with HNF4a LBD point mutants to explore unusual regulation of RARb gene expression by HNF4a. These biochemical approaches were straight forward and sound for data presentation. However, due to confusing nature of the controversial observations, the authors should address the following few issues and respond accordingly to gain an insight into a novel molecular mechanism and obtain more clear conclusions.   

Major issues

1.     The authors claimed that DR8 element in human RARb gene promoter is involved in HNFa response based on the observation with one deletion reporter gene construct. DR8 upstream half consensus DNA sequence mutation(s) should additionally be utilized to show more clear involvement.

2.     Suppression or activation of RARb gene through specific DNA regions by HNF4a was presented by reporter gene analysis only. To address whether the observed effect is mediated by direct DNA binding or through interaction with RAR/RXR, EMSA assay with RARE2 and immunoprecipitation analysis are recommended, especially because the HNF4a regulation with RARE in Epo and Gck genes are mediated through unusual DR2 elements not DR5.  

3.     In vitro confirmation of the observation presented in Fig. 1C is recommended in the cells without endogenous HNF4a or HepG2 cells after HNF4a gene silencing.

4.     By the same token, each RAR isoform specific gene silencing would be interesting for Cyp26a1 gene expression (endogenous or reporter gene) by RA treatment in the cells (maybe CV-1 or non-hepatic cell line) without endogenous HNF4a plus/minus HNF4a overexpression.

5.     The authors used various HNF4a LBD mutants to explain RA regulation of its transcriptional activity indirectly. If possible, direct RA binding to HNF4a or its interference with fatty acid occupancy has to be demonstrated with critical biochemical approaches such as FRET or AIMS (affinity isolation/mass-spectrometry). Fatty acid occupancy in HNF4a appears to be controversial for its transcriptional activity (Yuan et al., 2009).

Minor issues

1. Description of mouse Rarb-luc reporter shown in Fig. 2C is missing.

2. Related with major issue 1. Figure 2D mentioned in line 289 doesn’t exist. Probably additional data from sequence mutation study will fit to the missing Figure for the sentence.

3. The authors used a mouse model at the beginning of this study. Addition of mouse HNF4a amino acid sequence to the Fig. 5 would be ideal.

Author Response

Comments and Suggestions for Authors

The authors tested the expression of Cyp26a1 gene in the liver specific Hnf4a KO mice. Contrary to their earlier in vitro observation, Cyp26a1 mRNA level was increased in the HNF4a deleted liver especially after RA treatment. Upregulation of a nuclear hormone receptor Rarb by HNF4a deletion or RA treatment was attributed to the controversial Cyp26a1 gene expression by HNF4a. Therefore, the authors launched various in vitro analyses including reporter gene assays along with HNF4a LBD point mutants to explore unusual regulation of RARb gene expression by HNF4a. These biochemical approaches were straight forward and sound for data presentation. However, due to confusing nature of the controversial observations, the authors should address the following few issues and respond accordingly to gain an insight into a novel molecular mechanism and obtain more clear conclusions.   

Response: Thank you for your careful reading and thoughtful comments. Because of the limited time for revisions allowed by the journal, and because of recent retirements, we were unable to conduct additional studies. However we have addressed all the comments in terms of revisions to the text and have been careful not to overinterpret results. Again, thank you for an excellent review.

Major issues

  1. The authors claimed that DR8 element in human RARb gene promoter is involved in HNFa response based on the observation with one deletion reporter gene construct. DR8 upstream half consensus DNA sequence mutation(s) should additionally be utilized to show more clear involvement.

Response: Our speculation of DR8 involvement in activation of RARb promoter is that this site is located in the proximal promoter region with 2 other DR5 RARE which are necessary for transactivation of the RARb promoter. In fact, DR8 site is clustered with DR5 RARE2. We agree the upper half DR should be tested in the future. Thank you for your recommendation!  

  1. Suppression or activation of RARb gene through specific DNA regions by HNF4a was presented by reporter gene analysis only. To address whether the observed effect is mediated by direct DNA binding or through interaction with RAR/RXR, EMSA assay with RARE2 and immunoprecipitation analysis are recommended, especially because the HNF4a regulation with RARE in Epoand Gck genes are mediated through unusual DR2 elements not DR5.

Response: HNF4a binds not only to the DR2 elements present in Epo and Gck (Takako et al., 2001; Li et al., 2014 ) but also to GR5 (Zolfaghari & Ross, 2014).  To address this, we have added a sentence with references in the discussion section (line #’s 474 and 475).

  1. In vitro confirmation of the observation presented in Fig. 1C is recommended in the cells without endogenous HNF4a or HepG2 cells after HNF4a gene silencing.

Response: We have previously used the human embryonic kidney HEK293 cell line which lacks HNF4a expression (Zolfaghari & Ross, 2014). HNF4α significantly suppressed the inductive effect of RARs/RXRα on the promoter activity of the CYP26A1 gene in HEK293T cells following treatment with RA. That is what we have also observed in HepG2 cells.

  1. By the same token, each RAR isoform specific gene silencing would be interesting for Cyp26a1 gene expression (endogenous or reporter gene) by RA treatment in the cells (maybe CV-1 or non-hepatic cell line) without endogenous HNF4a plus/minus HNF4a overexpression.

Response: Previously we have shown that RAR isoforms including RARa and RARb highly induce the promoter activity of CYP26 in HepG2 cells after treatment with RA (see Zhang et al., 2010; Zolfaghari & Ross, 2014; Zolfaghari et al., 2019 and 2020). Although the CYP26A1 gene is highly expressed in HEK293T cells its expression is much less inducible in the cells treated with RA as compared to that in HepG2 cells. In HepG2 cells Cyp26A1 is barely expressed but when the cells are treated with RA its expression is induced over 1000-fold.

  1. The authors used various HNF4a LBD mutants to explain RA regulation of its transcriptional activity indirectly. If possible, direct RA binding to HNF4a or its interference with fatty acid occupancy has to be demonstrated with critical biochemical approaches such as FRET or AIMS (affinity isolation/mass-spectrometry). Fatty acid occupancy in HNF4a appears to be controversial for its transcriptional activity (Yuan et al., 2009).

Response: We do not know whether RA acts on the transactivation of HNF4a directly or indirectly. We have mentioned this throughout the manuscript. As you recommended, we would need to do FRET or AIMS experiments. Thank you for your suggestion for future studies.

On the other hand, fatty acids (FAs) were found as constitutively-bound endogenous embedded ligands for HNF4α by elegant crystallography studies of both in rat and human HNF4a

(Dhe-Paganon et al., 2002). We have discussed about the findings of this report throughout the results and discussion sections of this manuscript.

Minor issues

  1. Description of mouse Rarb-luc reporter shown in Fig. 2C is missing.

Response: We added a brief description in the legend of the Figure 2C.

  1. Related with major issue 1. Figure 2D mentioned in line 289 doesn’t exist. Probably additional data from sequence mutation study will fit to the missing Figure for the sentence.

Response: We corrected that to Figure 4C. Thank you!

  1. The authors used a mouse model at the beginning of this study. Addition of mouse HNF4a amino acid sequence to the Fig. 5 would be ideal.

Response: We added mouse HNF4a protein sequence in the alignment of the figure 5. Thank you for your recommendation.

Round 2

Reviewer 1 Report

The authors have updated the manuscript accordingly. 

No Concerncs

Reviewer 2 Report

Addressed to the raised issues properly.